# Methylation-Based Characterization of a New *IDH2* Mutation in Sinonasal Undifferentiated Carcinoma

**DOI:** 10.3390/ijms25126518

**Published:** 2024-06-13

**Authors:** Simon Burgermeister, Simona Stoykova, Fanny S. Krebs, Vincent Zoete, Martial Mbefo, Kristof Egervari, Antoine Reinhard, Bettina Bisig, Ekkehard Hewer

**Affiliations:** 1Department of Laboratory Medicine and Pathology, Lausanne University Hospital and University of Lausanne, 1011 Lausanne, Switzerland; simona.stoykova@chuv.ch (S.S.); martial.mbefo@chuv.ch (M.M.); bettina.bisig@chuv.ch (B.B.); 2Computer-Aided Molecular Engineering, Department of Oncology UNIL-CHUV, University of Lausanne, 1066 Epalinges, Switzerland; fanny.krebs@unil.ch (F.S.K.); vincent.zoete@unil.ch (V.Z.); 3Ludwig Institute for Cancer Research, 1066 Epalinges, Switzerland; 4Molecular Modelling Group, SIB Swiss Institute of Bioinformatics, 1015 Lausanne, Switzerland; 5Service of Clinical Pathology, Department of Diagnostics, Geneva University Hospitals, 1206 Geneva, Switzerland; kristof.egervari@hcuge.ch; 6Department of Otorhinolaryngology-Head and Neck Surgery, Lausanne University Hospital, 1011 Lausanne, Switzerland; antoine.reinhard@chuv.ch

**Keywords:** methylation analysis, molecular modeling, *IDH2* mutation

## Abstract

Mutations affecting codon 172 of the isocitrate dehydrogenase 2 (*IDH2*) gene define a subgroup of sinonasal undifferentiated carcinomas (SNUCs) with a relatively favorable prognosis and a globally hypermethylated phenotype. They are also recurrent (along with *IDH1* mutations) in gliomas, acute myeloid leukemia, and intrahepatic cholangiocarcinoma. Commonly reported mutations, all associated with aberrant IDH2 enzymatic activity, include R172K, R172S, R172T, R172G, and R172M. We present a case of SNUC with a never-before-described *IDH2* mutation, R172A. Our report compares the methylation pattern of our sample to other cases from the Gene Expression Omnibus database. Hierarchical clustering suggests a strong association between our sample and other IDH-mutant SNUCs and a clear distinction between sinonasal normal tissues and tumors. Principal component analysis (PCA), using 100 principal components explaining 94.5% of the variance, showed the position of our sample to be within 1.02 standard deviation of the other IDH-mutant SNUCs. A molecular modeling analysis of the *IDH2* R172A versus other R172 variants provides a structural explanation to how they affect the protein active site. Our findings thus suggest that the R172A mutation in *IDH2* confers a gain of function similar to other R172 mutations in *IDH2*, resulting in a similar hypermethylated profile.

## 1. Introduction

Sinonasal undifferentiated carcinoma (SNUC) is a rare and aggressive tumor arising in the nasal cavity and paranasal sinuses harboring a mutation in codon 172 of the isocitrate dehydrogenase 2 (*IDH2*) gene in the majority of reported cases [1]. Mutations of the *IDH1* and *IDH2* genes have been reported in a variety of malignancies, including acute myeloid leukaemias, gliomas, and cholangiocarcinoma. Somatic IDH mutations always occur in residues within the active site of those genes, with hotspots in one of three arginine residues critical for isocitrate binding (*IDH1* R132, *IDH2* R140, and *IDH2* R172) [2]. These mutations interfere with normal enzyme activity and catalyse isocitrate to 2-hydroxyglutarate (2-HG) instead of alpha-ketoglutarate (α-KG), thus leading to neomorphic enzyme activity [3,4]. 2-HG accumulation then leads to the inhibition of histone demethylases and TET family 5-methylcytosine hydroxylases, producing a global DNA hypermethylation. Commonly reported mutations of codon 172 of the *IDH2* gene include R172K, R172S, R172T, R172G, and R172M [3]. In this article, we present a case of SNUC with a never-before-described *IDH2* mutation, R172A. Although the substitution affects the same codon as other previously described mutations, it is not uncommon for different single-amino acid replacements affecting the same site to induce widely different consequences on protein function. Examples include R248W vs. R248Q mutations of *TP53* [5,6], and G12D vs. G12V mutations of *KRAS* [7]. It is therefore relevant to characterize the effect of a newly discovered mutation, even though different mutations occurring on the same codon have been frequently described. Given that all previously reported mutations affecting codon 172 of *IDH2* are associated with neomorphic enzyme activity, ultimately leading to hypermethylation, in this study, we investigated this possible scenario for our sample using Illumina Methylation Arrays. The findings were then compared to similar data extracted from the Gene Expression Omnibus (GEO) database.

## 2. Results and Discussion

### 2.1. Case Description

A 48-year-old non-smoker male patient presented with nasal obstruction for over a year. A computed tomography (CT) scan and magnetic resonance imaging (MRI) revealed a large, infiltrating mass involving the left nasal cavity (Figure 1) and ethmoid sinus, extending into the periorbital fat and brain parenchyma at the level of the right gyrus. Biopsy and histopathologic examination led to the diagnosis of SNUC (Figure 2a) with increased H3K27 trimethylation (Figure 2b), compatible with the hypermethylated phenotype. It was classified as stage cT4b based on the radiological findings. Molecular profiling on the biopsy sample showed a previously not reported mutation of *IDH2* (R172A) and a common *TP53* mutation (R273H) [8,9]. A positron emission tomography (PET) using [^18^F]fluorodeoxyglucose ([^18^F]FDG) highlighted an intense hypermetabolism of the lesion (Figure 3) without evidence of metastasis. Following the diagnosis, the patient underwent three cycles of neoadjuvant chemotherapy composed of a combination of docetaxel, carboplatin, and 5-fluorouracil (TCF). The lesion was thereafter surgically removed following 3 months of treatment. The pathological examination of the surgical specimen showed important post-treatment remodeling, indicating a good response to chemotherapy and showing only a few residual tumor cells in the nasal septum, spanning 0.6 mm on its longest axis (ypT1). No evidence of loco-regional or distant metastasis was found.

### 2.2. Molecular Analysis

Targeted NGS revealed a known pathogenic variant c.818G>A (p.R273H) in exon 8 of the *TP53* gene, at a variant allele frequency (VAF) of 76%. Another yet-to-be-classified variant, c.514_515delinsGC, encoding the substitution of Arginine to Alanine at position 172 (p.R172A), was identified in exon 4 of *IDH2*, with a VAF of 40%. The latter mutation is not registered in any of the databases we explored (COSMIC [10], cBioPortal [11], ClinVar [12], OncoKB [13], and Jax-CKB [14]) and is not reported in the literature. This mutation alters a known hotspot with an Arginine residue critical for isocitrate (ICT) binding [2], located in a highly conserved region (Figure A1).

IDH2 is a mitochondrial enzyme that plays a crucial role in cellular metabolism and is involved in the Krebs cycle, which is a central metabolic pathway that occurs in the mitochondria. IDH2 allows the conversion of NADP+ into NADPH in the presence of D-threo-isocitrate. Several IDH2 experimental structures are available in the Protein Data Bank [15] (PDB). However, no human wild-type experimental structure showing the position of residue R172, or the substrate in the protein active site, exists. An experimental structure of IDH2 from mice, co-crystallized in the presence of ICT and Mg^2+^, has been released. The IDH2 of Homo sapiens and Mus musculus are highly similar, with a sequence identity of 95%. Both are composed of 452 amino acids, with the arginine of interest in position 172 (Figure 4). Of note, all residues in the vicinity of R172 and the substrate are conserved. The residues that differ between the two organisms are predicted to not affect the active site and the substrate binding regions’ structural conformations. The high similarity between the two proteins in these organisms enables the structural analysis of IDH2 using experimental murine structures. Thus, the murine structure with the PDB ID 5h3f [16] was used in this study, after renumbering the residues based on the human sequence for easier discussions.

IDH2 is a homodimer protein (Figure 5a). R172 is situated within the protein active site, more precisely within the ICT anchoring region, and participates in various interactions with the ICT substrate and neighboring residues. Arginine is a positively charged, large, and amphiphile residue, capable of interacting with both polar and nonpolar functions. This allows R172 to participate in (i) hydrophobic interactions with V147, A174, and I170; (ii) hydrogen bonds with N310 and ICT; (iii) salt bridges with ICT and D314; and (iv) a cation-π interaction with Y179 (Figure 5b). R172, therefore, plays an important role in IDH2 structural stability around the active site. Noticeably, ICT chelates to the Mg^2+^ on one side, while two of its carboxylate functions make ionic interactions with three highly conserved arginines at positions 140, 149, and 172 on the other side. It seems that this specific interaction scheme is crucial to enable optimal enzymatic activity, as mutations of R140 and R172 are known to decrease the ICT conversion into α-KG and increase the conversion of α-KG into R-2-hydroxyglutarate, which is known as an onco-metabolite [20,21,22]. Of note, no mutation involving R149 has been characterized yet.

Structural analysis reveals that replacing any of the three arginine residues is not feasible without affecting the protein structure in the vicinity, as both the size and positive charge of these residues play crucial roles. Even a positively charged amino acid like Lysine falls short in reproducing arginine interactions due to its smaller size. Mutations of R172 to lysine, glycine, methionine, and serine [23,24,25] lead to a gain of function. A structural model was created for each of these mutations, including the mutation of interest (*IDH2* p.R172A), to analyze their impact on the molecular interactions existing in the wild-type system. All models were generated with the software FoldX version 5 [26], which was designed to assess and forecast the impact of mutations on protein stability and binding affinity.

Examining the generated models highlights the importance of both the size and the charge of the wild-type residue (Figure 6), which are not reproduced by Alanine. The *IDH2* p.R172A mutation is predicted to have an important structural impact as it can only reproduce a few wild-type hydrophobic interactions with V147, A174, and I170. The smaller the mutated residue, the greater the impact on the structural stability (p.R172G/A/S, Figure 6b–d). Alternatively, mutated residues of larger size could stabilize the region via stronger hydrophobic interactions (p.R172M/K, Figure 6e,f). Nevertheless, these mutated residues cannot interact with the substrate, which may affect the binding of the latter. In summary, none of the generated mutants, including p.R172K, can maintain the optimal environment for the enzymatic reaction observed in the wild-type protein.

### 2.3. Methylation Analysis

Brain classifier v11b4 [27] could not classify our sample using the Illumina array. It was, however, classified as *IDH2*-mutated with a calibrated score of 0.99 using the newer Brain Tumor classifier v12.5 [28], which has this tumor class in its reference, while the v11b4 version does not. Hierarchical clustering using the 10,000 most differentially expressed probes on the methylation data arrays showed the proximity of our sample to other *IDH1/2* mutant SNUCs (Figure 7).

These most differentially expressed probes were thus enough to differentiate *IDH1/2* mutant lesions and cell lineage. They were predominantly located in gene body regions, a pattern that follows the same proportion as in the overall probes kept for analysis (Table 1).

Other *IDH1/2* mutant samples showed clustering according to pathology findings. Interestingly, however, this did not differentiate clearly between normal sinonasal tissue and the squamous cell carcinoma of the sinonasal region, suggesting a lack of specific methylation patterns in the squamous cell carcinoma of the sinonasal region. *IDH1/2* and *SMARCB1*-deficient sinonasal carcinoma, however, both show a clear differentiation, indicating significant methylation changes compared to normal samples of the same anatomical region. We also observed (Figure 8) that although *IDH1/2* mutations are shared among several samples, they still cluster predominately according to the tissue of origin, as previously reported [29]. This suggests that *IDH1/2* mutations act on a methylation background and that the overall methylation profile remains highly dependent on cellular origin. In utilizing the first 100 components, PCA explained 94.5% of the variance and positioned our sample within a 1.02 standard deviation (SD) of the other *IDH1/2* mutant SNUC cases.

From the analyses, we see that our sample displays a methylation pattern similar to other IDH-mutant SNUCs. Given that the methylation phenotype of the IDH-mutant samples is determined, among others, by the accumulation of 2-HG produced through neomorphic enzyme activity of the *IDH1/2* alleles, this suggests that the R172A mutation on *IDH2* confers a gain of function similar to other reported *IDH1/2* mutants in SNUC. Interestingly, the position of the samples in the principal components space is also highly dependent on the cell population origin. This would indicate that, although the *IDH1/2*-mutated samples show an increased methylation pattern, the background methylation state associated with cell lineage differentiation is still determinant. The cellular origin of lesions can therefore still be classified using methylation data, even if they have mutations widely increasing the overall methylation levels. The findings of our study come with some limitations. The fact that this mutation is found in only one individual lesion is the most obvious. However, given that the mutation had never been reported previously, this is, to our knowledge, the only available sample, and we still believe it is worth reporting the findings associated with it. Additionally, although the external datasets from the GEO database offer an important insight into methylation patterns according to tissues of origin and lesion types, we could not obtain additional information on which of the isocitrate dehydrogenase was affected in mutated samples (*IDH1* or *IDH2*), limiting the precision of our analysis. This point should be assessed in further studies with access to more comprehensive datasets. Furthermore, as the external data come from different institutions, it is possible that differences in sample preparation could influence measurements provided by the methylation arrays. However, the clustering analysis could not find any such trend, and other studies [30] have also successfully used some of the same data from several labs without any obvious biases. However, this possible limitation should be mentioned and possibly further investigated if methylation-based analysis becomes more widespread in the future. It is also important to highlight that due to the treatment of the viable tissue and the available techniques in our institute, the level of 2-HG could not be measured for this case. Thus, the 2-HG levels can only be supposed to be increased based on increased overall methylation levels, the observed methylation pattern, and molecular modeling but could unfortunately not be confirmed empirically. Finally, a plurality of most differentially expressed probes, which are, therefore, the most determinant for clustering, were located on gene body regions. The regulatory effect of DNA methylation in these regions is inadequately understood, and interpretation is therefore limited.

## 3. Materials and Methods

Sequencing and methylation analyses were performed on tumor DNA extracted from the formalin-fixed paraffin-embedded biopsy samples. Targeted next-generation sequencing (NGS) was carried out using a custom amplicon-based panel covering the hotspots of 52 cancer-related genes, including *IDH1* and *IDH2* [31]. Sequencing was performed on the Ion GeneStudio S5 System (Ion Torrent, Thermo Fisher Scientific, Waltham, MA, USA). Immunohistochemistry for H3K27me3 (clone C36B11) was performed on the Ventana Benchmark Ultra platform (Ventana Medical Systems Inc., Tucson, AZ, USA) using a Ventana Optiview detection kit. The human and mouse IDH2 protein sequences were retrieved from UniProt [17]. Sequence alignments were performed using Clustal Omega v.1.2.4 [18]. UCSF Chimera v.1.17.3 was used for the structural and sequence visualizations and analysis [19]. Models were generated using FoldX v.5 [26]. The analysis of methylation sites was performed using the Infinium Methylation EPIC BeadChip array from Illumina (San Diego, CA, USA). The methylation data were loaded on the Brain Tumour classifier from Heidelberg University. Our sample was also compared with others from the literature, including other sinonasal tumors such as squamous cell carcinoma, *SMARCB1*-deficient sinonasal carcinoma (SNC), normal sinonasal tissues, and other *IDH1/2* mutated tumors. Details on the external data are available in Table A1.

Methylation data preprocessing was performed using R version 4.2.2. Raw methylation data were preprocessed with the minfi [32] package, which was used to subset overlapping probes between 450 K and 850 K arrays using the combineArrays function. The same package was used to verify the quality of the samples by calculating the mean detection *p*-values. Sex probes were removed. For array normalization, preprocessSWAN was chosen to correct them for differences between type I and II probes. M-values were extracted for each sample and used for further analysis, as they have been shown to be more statistically valid than the Beta-value for the differential analysis of methylation levels [33]. Differential analysis was performed in Python v3.8.10 using SciPy [34] (version 1.9.3) and statsmodel [35] (version 0.14). Hierarchical clustering was performed using only the 10,000 most differentially methylated probes, which maximized the standard deviation. Principal component analysis (PCA) was performed on all probes from M-values using scikit-learn [36]. Uniform Manifold Approximation and Projection (UMAP) was performed on M-values using umap-learn [37]. Probes were separated into gene body regions, promoter regions (TSS200 & TSS1500), and enhancer regions using the UCSC gene region information provided for each probe by Illumina. For visualization, we used the Matplotlib [38] and Seaborn [39] libraries.

## 4. Conclusions

This work demonstrates the evaluation of new mutations from a readily available methylation array compared to results freely available from external databases using computational tools. Based on the results obtained from this sample, our findings suggest that the DNA methylation pattern associated with the R172A mutation is consistent with the already described *IDH1/2*-mutant SNUCs. The findings are therefore consistent with R172A mutation on *IDH2* conferring a gain of function similar to other reported *IDH2* mutants in SNUC. This should, however, be investigated further if other such samples become available in the future. We also observed that IDH-mutant SNUCs form a distinct methylation cluster, different from the methylation pattern observed in other *IDH1/2*-mutated tumors, consistent with IDH-mutated samples remaining highly dependent on the cell origin. In using a methylation array, samples can therefore still be clustered according to cellular origin even for those harboring IDH mutations, which significantly affect methylation levels. This highlights the promising perspective of methylation and molecular modelization analysis for diagnostic and phenotype assessment.

## Figures and Tables

**Figure 1 ijms-25-06518-f001:**
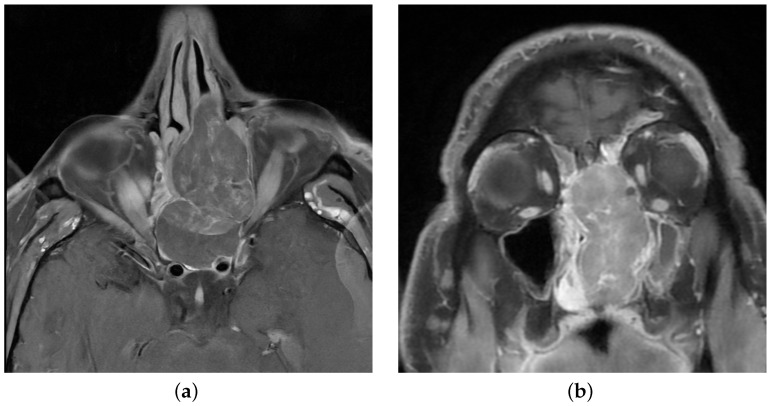
Magnetic resonance imaging showing a heterogeneous mass in the left nasal cavity. (**a**) T1 with Gadolinium, axial view. (**b**) T1 with Gadolinium, coronal view.

**Figure 2 ijms-25-06518-f002:**
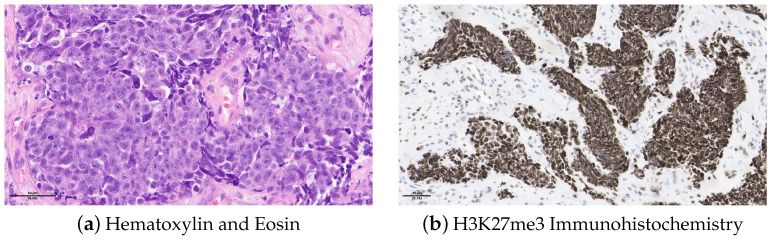
Histopathology of the biopsy sample. (**a**) H&E showing sheets of epithelioid cells without any definite lineage differentiation. The nuclei are medium- to large-sized and hyperchromatic. We note the presence of tumor necrosis and frequent mitosis. (**b**) H3K27me3 IHC showing an increased labeling for our sample compatible with increased histone H3K27 trimethylation.

**Figure 3 ijms-25-06518-f003:**
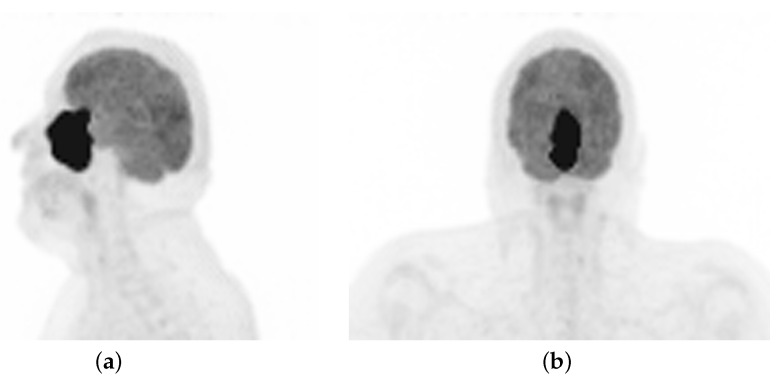
[^18^F]FDG PET maximum intensity projection showing a large, irregular and intensely hypermetabolic mass in the sinonasal region. (**a**) lateral view. (**b**) frontal view.

**Figure 4 ijms-25-06518-f004:**
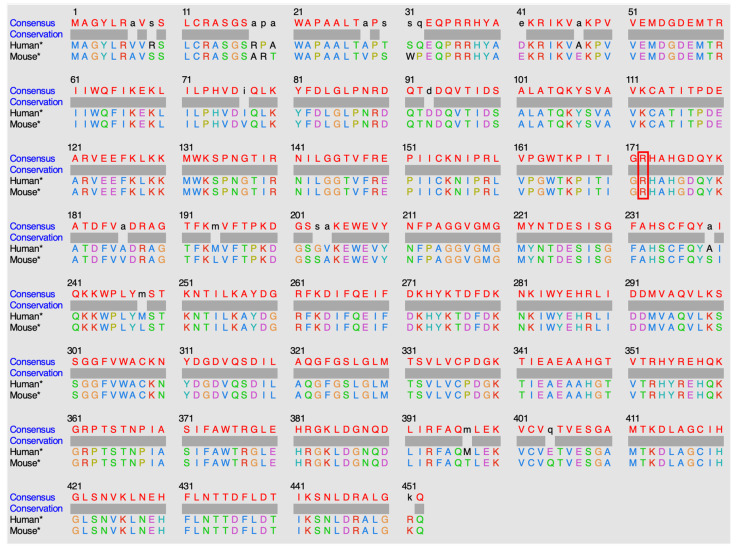
Alignment of human and mouse protein sequences. Sequences retrieved from UniProt database [17], alignment performed with Clustal Omega v.1.2.4 [18], and sequences visualized with UCSF Chimera v.1.17.3 [19]. The color code of the alignment corresponds to the default colour code of the visualization tool. The star (*) indicates that the canonical IDH2 sequence of the organism has been used. The red rectangle indicates the position of the amino acid 172 in humans.

**Figure 5 ijms-25-06518-f005:**
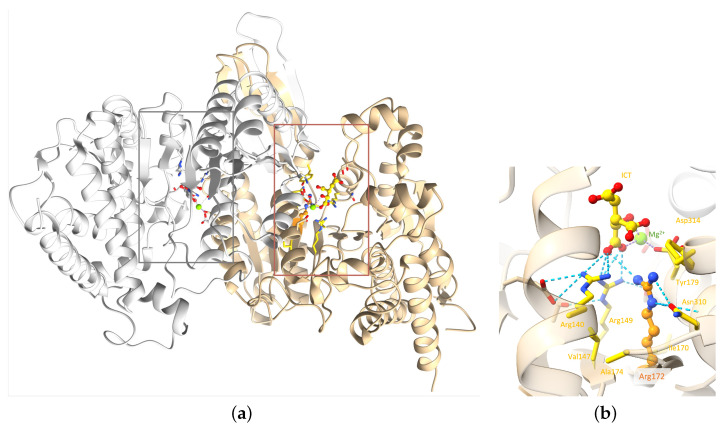
IDH2 3D homodimeric structure. (**a**) The IDH2 homodimer with the two active sites indicated by thin boxes; (**b**) zoom in on R172 of chain A of the complex. Chains A and B are colored tan and gray, respectively. R172 and ICT are shown as balls and sticks: Mg^2+^ as a green sphere and important residues as sticks. R172 is colored orange. Elements interacting directly with R172 are colored yellow. Hydrogen bonds involving the arginine 140, 149, and 172 are represented as blue dashed lines. For easier visualization, the protein is represented by transparent ribbons. (PDB ID: 5h3f [16]).

**Figure 6 ijms-25-06518-f006:**
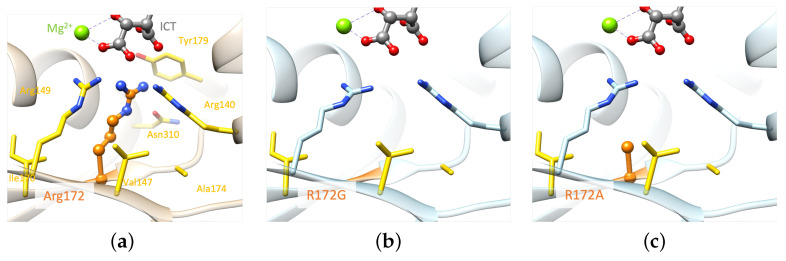
Comparison of structural models of the mutated proteins with the wild-type IDH2 active site. (**a**) Wild-type; (**b**) structural models of IDH2 p.R172G, (**c**) IDH2 p.R172A, (**d**) IDH2 p.R172S, (**e**) IDH2 p.R172M, and (**f**) IDH2 p.R172K. Only monomer A is represented for clarity. Residue 172 and ICT molecule are shown as balls and sticks: Mg^2+^ as a green sphere and important residues as sticks. Residue 172 is colored orange. Elements interacting directly with residue 172 are colored yellow. For easier visualization, the protein is represented by transparent ribbons. The dashed lines represent the interactions between ICT and Mg^2+^. Models were built based on the experimental structure with the PDB ID 5h3f [16].

**Figure 7 ijms-25-06518-f007:**
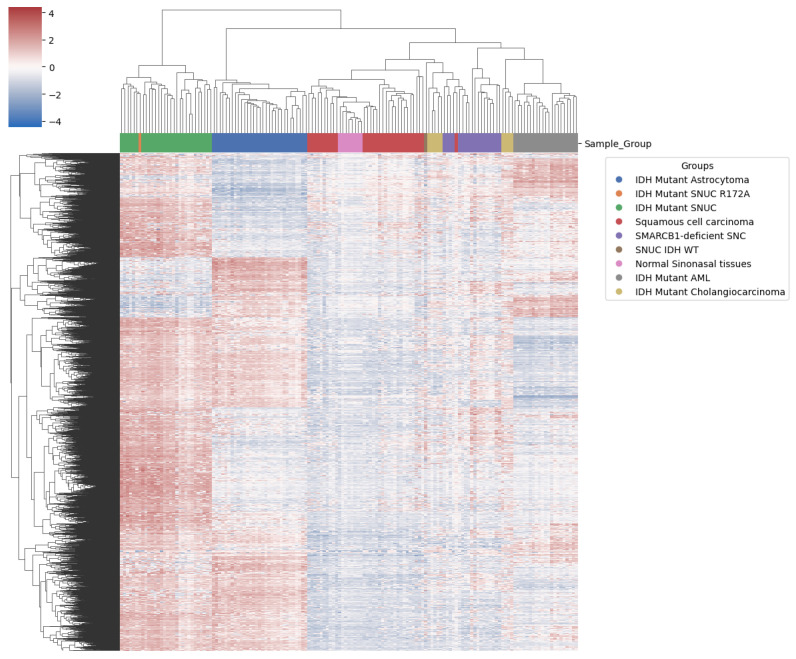
Heatmap with clustering using top most differentially methylated probes. Our sample (in orange) shows proximity to conventional IDH mutant SNUC (green).

**Figure 8 ijms-25-06518-f008:**
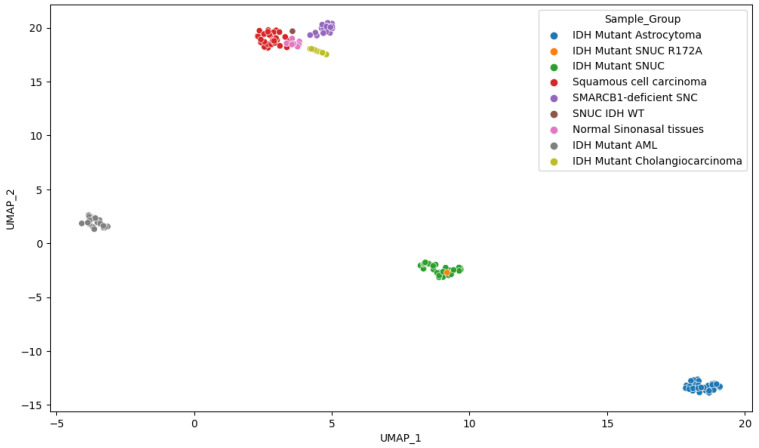
UMAP representation of the M-values.

**Table 1 ijms-25-06518-t001:** Probes region distribution among the 10,000 most expressed ($N_{10k}$) and overall array post preprocessing selection ($N_{overall}$).

Regions	$N_{10k}$	%	$N_{\mathrm{overall}}$	%
TSS200	1237	12.37	50,040	13.51
TSS1500	1643	16.43	64,419	17.39
5UTR	1482	14.82	50,649	13.67
Body	3805	38.05	134,277	36.25

## Data Availability

The data that support the findings of our study are available from the corresponding authors upon reasonable request.

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
