# Peer review of "Methylation-Based Characterization of a New IDH2 Mutation in Sinonasal Undifferentiated Carcinoma"

_ijms, 2024, doi:10.3390/ijms25126518_

Round 1

Reviewer 1 Report

Comments and Suggestions for Authors

The authors have identified an example of a tumor, specifically a sinonasal undifferentiated carcinoma (SNUC), from a patient that carries a previously undescribed mutation in the isocitrate dehydrogenase gene 2 (IDH2).  Mutations of the IDH1 and IDH2 genes have been reported in a variety of malignancies. IDH2 is a mitochondrial enzyme involved in cellular metabolism and the Krebs cycle. Mutations in IDH2, including at codon 172, have been seen in multiple types of cancers. SNUC is a rare tumor that, in previously reported instances, carries multiple mutations of the IDH2 gene including a mutation in codon 172. Here, the specific mutation results in an arginine being replaced with alanine (R172A). This is the first description of R to A substitution at that location.

The authors have described the clinical case from which the tumor originated, including images of the tumor mass (Figure 1), and used positron emission tomography using [18F]fluorodeoxyglucose to demonstrate intense hypermetabolism associated with the tumor (high level of glucose uptake) (Figure 2). Histopathology and immunohistochemistry reveal the undifferentiated structure of the tumor tissue and the increased presence of histone hypermethylation in the tissue (Figure 3). IDH2 of human and mouse are highly similar, with a sequence identity of 95%. Both are composed of 452 amino acids, with the arginine of interest in position 172 (Figures 4 and 5). Because no human wild-type crystal structure is available showing the position of residue R172 , or substrate in the protein active site, the authors used the available structure of IDH2 from mice, for modeling potential structural changes associated with the R172A mutation (Figures 6 and 7). Modeling studies result in the conclusion that the alanine substitution affects interactions in the region and loss of integrity of the enzyme active site. Methylation site data from public databases were compared across other SNUCs and other tumor types, and demonstrated a clustering of patterns among the IDH mutant SNUCs (Figures 8 and 9).

The findings are consistent with expected changes in the tumor metabolic profile and level of histone hypermethylation. This is an apparently unique mutation with downstream consequences consistent with previously reported findings of similar mutations. The justification for the work is laid out logically, the methods are appropriate, and the conclusions align with the data presented. Overall, a very good and very interesting paper.

Author Response

We thank the reviewer for his detailed and thoughtful review of our manuscript. We are pleased to hear that he found our manuscript interesting and we appreciate his positive feedback and recognition of the significance of our findings.

Reviewer 2 Report

Comments and Suggestions for Authors

In manuscript ijms-3001370, the authors present a case report of a novel mutation in the IDH2 gene found in a man affected by sinonasal undifferentiated carcinoma (SNUC). The authors describe the mutation, its associated phenotype in terms of clinical and molecular (DNA methylation) characterization, and a prediction of the structural modification of the protein, caused by the mutation itself. The mutation affects a specific amino acid that is already known to be mutated in SNUC, although previously described mutations are different from the present one (alanine here, other amino acids elsewhere). In the introduction (lines 33-36), the authors correctly note that it is important to characterize single mutations also in mutation hot spots, as it is possible that different amino acid substitutions bring to different phenotypes. Unfortunately, this is not the case, as the DNA methylation pattern in the patients is comparable (lines 138 and following) to that described in other patients with different mutations in the same position. Thus, the novelty of the report merely relies on the presence of an amino acid substitution not described before (R172A), that nonetheless (i) falls in a very well-known position (lines 28-29), (ii) is associated to the same pathology already described in literature (i.e., SNUC) (lines 45-47), and (iii) causes an alteration of the DNA methylation pattern widely overlapping that of other described mutations in the same hotspot (lines 138-139).

In addition to the poor novelty of the data in the manuscript, other major issues are present.

1.      Together with R172A in IDH2, the authors also report the finding of the mutation R273H in TP53 (lines 48-50). Beyond the lack of references and description of this mutation in human cancers (examples, but there are many others: doi: 10.1007/s10585-007-9084-8; doi: 10.1038/s41467-022-30481-7; doi: 10.1080/07391102.2023.2283793; doi: 10.1158/1541-7786.MCR-22-0133), the authors do not report any investigation on this second mutation, which sounds strange (i) in consideration of the role of TP53 in human cancers, including sinonasal ones, and (ii) in consideration of what said before about the R172A mutation in IDH2 and its low impact on the advancement of knowledge about SNUCs.

2.      The accumulation of 2-HG in this specific sample was not verified, but only deduced (lines 141-142) by similarity with other IDH2 mutations described in literature. In this perspective, a molecular characterization of the functional role of R172A mutation is missing in the manuscript.

3.      Deductions on the structural role of R172A mutation in the protein conformation come from the mouse model, that indeed is 95% identical, but nonetheless is another protein. Without (at least) the data about 2-HG accumulation, this analysis is only a speculation on the role of R172A in man.

4.      Introduction should include information and related references about TP53 role in SNUC in particular, and sinonasal cancers in general, as TP53 mutation is an integral part of the genotype of the affected patient and likely involved in his clinical condition.

Other minor points:

1.      Reference in line 133 is badly formatted.

Author Response

Following are our responses to the points addressed by the reviewer in the order they were raised:

1) We thank the reviewer for this remark. Indeed, the TP53 mutation described in this case was lacking context and citation. This has been corrected with the introduction of citations on line 51.

We must note, however, that our primary interest was the IDH2 mutation first because of its novelty and also because IDH1/2 mutations define a clinically meaningful subgroup in SNUC that can be investigated through methylation analysis which is not the case for TP53 mutations.

While it is true that it doesn’t add to the overall understanding of SNUCs, we feel it is still worth reporting. Especially given the fact that different mutations affecting the same codon have been found to lead to widely different phenotype as is mentioned in the introduction. In the specific case of IDH2, the observed hypermethylated phenotype is dependent on neomorphic enzyme activity but other not reported mutations could lead to the inactivation of the enzyme or similar enzyme activity to the wild type of variant, both cases would likely not lead to a hypermethylated phenotype. While our research does not provide definitive proof, it none the less an interesting observation that we feel is worth reporting.

2) This is an important point, and we thank the reviewer for raising it. Indeed the 2-HG accumulation in this sample cannot be confirmed with the current analysis. Furthermore, as the samples available for this analysis are all very limited in available viable material and consist only of formaldehyde fixed and paraffin-embedded (FFPE) tissues, for which there is, to our knowledge, no clear protocol or guidelines for such measurement on this kind of preprocessed tissues. This makes the interpretation of the measurements for one single sample difficult.

However, since molecular modeling predicts a similar configuration than for other known codon 172 mutations of IDH2 and bioinformatics analysis shows a similar methylation pattern, we feel that hypermethylation through increased levels of 2-HG is at least a very likely scenario that must be raised.

Since we cannot at present confirm it, however, we added this point to the limitation in the discussion (lines 168-172).

3) We agree with the reviewer that sequence identity information is not sufficient to transpose structural analyses from one organism to another, even if the sequence identity value is very high, as in our case. This is why we also performed a comparison of the active site region of the two proteins. We observed that the main residues that differ between the two organisms do not affect the position of interest or substrate binding. We thank the evaluator for highlighting that this was not described enough in the manuscript. We have now clarified this section.  

Previous version: “Both are composed of 452 amino acids, with the arginine of interest in position 172 (Figure 5). Of note, all residues in the vicinity of R172 and the substrate are conserved. The high similarity between the two proteins in these organisms enables structural analysis of IDH2 using experimental murine structures”

New version: “Both are composed of 452 amino acids, with the arginine of interest in position 172 (Figure 5). Of note, all residues in the vicinity of R172 and the substrate are conserved. The residues that differ between the two organisms are predicted to not affect the active site and the substrate binding region structural conformations. The high similarity between the two proteins in these organisms enables structural analysis of IDH2 using experimental murine structures”

4) IDH2 mutations SNUC have often been associated with other mutations, the most common of which is indeed p53 mutation (in 60% of IDH2-mutated SNUC according to doi: 10.1038/s41379-019-0285-x). However, there is no indication that mutation of TP53 affect the methylation patterns in SNUC. The overall hypermethylation status in our case is therefore more likely associated with mutation of IDH2. The effect of TP53 on response and survival in SNUC is equally unclear. While there is a clear indication that such mutations are an important prognostic factor in chronic lymphocytic leukemia and breast cancer, to our knowledge this has not been investigated for SNUC.

5) We thank the reviewer for this notification, the mistake has been corrected in the new version of the manuscript with the position being now on line 138. A similar badly formatted reference on line 205 was also corrected.

Reviewer 3 Report

Comments and Suggestions for Authors

Dear Authors,

please address the following issues.

- The manuscript is presented as "Article" but should be defined a "Case report".

- Please, be consistent with the gene names. All in italics (e.g. IDH1 and IDH2).

- Figure 4 and figure 5: please report the source/reference in the figure legend. 

- Lines 102-104: please, specify how the structural model for each mutation was produced. For instance, the name of the informatics tool or portal used to generate the tri-dimensional structures (I suppose is FoldX v.5 that you mention in the Mat&Met).

- Line 125: the choice of SMARCB1 mutations should be better specified.

- Title of chapter 3: Results and Discussion.

- Conclusion: I think that some statements must be improved. It should be specified that the case reported in the manuscript is one, therefore the fact that the R172A mutation clusters with the methylation profiles of SNUCs is limited to this case report. I am not saying that it could not be generalized in a future research, but I think the conclusion is too preliminary. 

The modelling and methylation datasets analysis are correct and interesting while the conclusions are limited. Figure 4 might be moved to a Supplementary material.

Comments on the Quality of English Language

English is fine.

Author Response

1) This is an important remark.  However, given that we also analyzed the methylation patterns of 154 other samples from the GEO database, we felt it was more suited to an article category. This also seemed to be the case for other similar publications in this journal (such as doi.org/10.3390/ijms21218021). However, if the editor feels it should be defined as a case report, we won’t object to this decision.  

2) We thank the reviewer for this information, we have now reviewed the manuscript with particular attention to this point.

3) Indeed, this was missing in the main text.  We have now modified the figure captions to provide this information. 

4) Indeed, this was missing in the main text. We thank the reviewer for mentioning this point. We have now modified the text to provide this information.

Previous version: (lines 101-103): “A structural model was created for each of these mutations, including the mutation of interest (IDH2 p.R172A), to analyse their impact on the molecular interactions existing in the wild-type system.”

New version: “A structural model was created for each of these mutations, including the mutation of interest (IDH2 p.R172A), to analyse their impact on the molecular interactions existing in the wild-type system. All models were generated with the software FoldX version 5 [Schymkowitz, J.; Borg, J.; Stricher, F.; et al. The FoldX web server: an online force field. Nucleic Acids Res. 2005, 1, W382–8. (the number 27 in the original version], which is designed to assess and forecast the impact of mutations on protein stability and binding affinity.”

5) Indeed, this is an important point, and we thank the reviewer for bringing it to our attention. We wanted to include examples of SWI/SNF complex-deficient sinonasal carcinoma and SMARCB1-deficient sinonasal carcinoma were the ones we could find on the GEO database. This has now been included in the method section (lines 168-172).

6) Indeed, this was a mistake, it has now been corrected, we thank the reviewer for bringing it to our attention.

7) We agree that the conclusion should have highlighted this important limitation as well as the discussion. The conclusion has now been changed to reflect this limitation.

8) We thank the reviewer for this good remark, and we have done this as suggested.

Round 2

Reviewer 2 Report

Comments and Suggestions for Authors

No additional comments